# Melinacidin-Producing *Acrostalagmus luteoalbus*, a Major Constituent of Mixed Mycobiota Contaminating Insulation Material in an Outdoor Wall

**DOI:** 10.3390/pathogens10070843

**Published:** 2021-07-04

**Authors:** (Aino) Maria A. Andersson, Johanna Salo, Raimo Mikkola, Tamás Marik, László Kredics, Jarek Kurnitski, Heidi Salonen

**Affiliations:** 1Department of Civil Engineering, Aalto University, P.O. Box 12100, FI-00076 Aalto, Finland; johanna.72salo@gmail.com (J.S.); raimo.mikkola@aalto.fi (R.M.); jarek.kurnitski@aalto.fi (J.K.); heidi.salonen@aalto.fi (H.S.); 2Department of Microbiology, Faculty of Science and Informatics, University of Szeged, Közép fasor 52, H-6726 Szeged, Hungary; mariktamas88@gmail.com (T.M.); kredics@bio.u-szeged.hu (L.K.); 3Department of Civil Engineering and Architecture, Tallinn University of Technology, Ehitajate tee 5, 19086 Tallinn, Estonia; 4International Laboratory for Air Quality and Health, Queensland University of Technology, 2 George Street, Brisbane, QLD 4001, Australia

**Keywords:** melinacidin, *Acrostalagmus luteoalbus*, guttation droplets, mixed mycobiota, indoor dust, outdoor wall, cork liner

## Abstract

Occupants may complain about indoor air quality in closed spaces where the officially approved standard methods for indoor air quality risk assessment fail to reveal the cause of the problem. This study describes a rare genus not previously detected in Finnish buildings, *Acrostalagmus*, and its species *A. luteoalbus* as the major constituents of the mixed microbiota in the wet cork liner from an outdoor wall. Representatives of the genus were also present in the settled dust in offices where occupants suffered from symptoms related to the indoor air. One strain, POB8, was identified as *A. luteoalbus* by ITS sequencing. The strain produced the immunosuppressive and cytotoxic melinacidins II, III, and IV, as evidenced by mass spectrometry analysis. In addition, the classical toxigenic species indicating water damage, mycoparasitic *Trichoderma, Aspergillus* section *Versicolores, Aspergillus* section *Circumdati, Aspergillus* section *Nigri*, and *Chaetomium* spp., were detected in the wet outdoor wall and settled dust from the problematic rooms. The offices exhibited no visible signs of microbial growth, and the airborne load of microbial conidia was too low to explain the reported symptoms. In conclusion, we suggest the possible migration of microbial bioactive metabolites from the wet outdoor wall into indoor spaces as a plausible explanation for the reported complaints.

## 1. Introduction

The term “microbiota” refers to the microbial community in a defined environment. The term “microbiome” refers to the total genome of such microbiota. In buildings, the colonizing microbiota and the microbiome are global and uniform compared to those outdoors, which are local and diverse [1,2]. In urban environments in high-income countries, the major microbial exposure by humans over their lifetime is to uniform building microbiota [3,4,5]. An indoor lifestyle leaves occupants at the mercy of uniform building microbiota, where microbial exposomes trapped indoors can reach higher concentrations and persist longer than those outdoors [1,3,6,7,8]. Microbes including *Aspergillus, Penicillium, Trichoderma, Fusarium, Chaetomium, Streptomyces, Bacillus,* and *Nocardiopsis* species, which produce bioreactive metabolites such as mycotoxins [9,10,11,12,13,14,15], immunoreactive substances [16,17,18], mitochondrial and ionophoric toxins, and fungicides and antibiotics, contribute to the building exposome in wet buildings worldwide [19,20,21,22,23,24,25,26,27,28,29,30,31,32,33].

The loss of biological diversity promotes dysbiosis, imbalances in the microbial ecosystem on or within the body, and loss of tolerance to environmental microbes [34,35,36,37,38,39,40,41,42,43]. Dysbiosis decreases resilience against changes in environmental exposure and promotes systemic subclinical inflammation, allergy, and asthma [44,45,46,47], symptoms also associated with wet buildings [48,49,50,51,52,53,54,55,56,57,58]. The pathophysiological mechanism behind the morbidity associated with wet buildings is still an open question [59,60,61,62,63], but enhancement of dysbiosis, loss of tolerance to environmental microbes, and activation of inflammasomes by exposure to immunoreactive antimicrobial substances from wet, “moldy” buildings cannot be excluded [13,35,44,51,54,56,57,58,64].

Recently, new indoor toxigenic fungi and their toxins have been identified using toxicity screening of culturable indoor fungal isolates [22,65,66]. Most metabolically active fungi found in indoor environments are detectable by cultivation [1,67]. Because fungi digest before they ingest, their metabolites are secreted into the environment. The secreted metabolites, including enzymes, signaling molecules, surfactants, etc., consist of much more than conidia and hyphal fragments [68,69]. New mechanisms of secretion and emissions of bioreactive substances into indoor air have been proposed, stressing the relevance of the viability and metabolic state of indoor microbes [6,70,71,72].

Occupants may complain about the indoor air quality in closed spaces where there are no visible signs of microbial growth and the airborne load of microbial particles is too low to explain the reported symptoms [1,73]. This study is a continuation of earlier studies in 2014–2020 concerning problems related to indoor air in a public building in Finland [65,66,71,73]. The aim of this study was to develop methods for elucidating the possible migration of secreted microbial metabolites from outer walls to offices with regard to indoor-related health complaints.

## 2. Results

### 2.1. Building Inspection Connected to Five Problematic Rooms

Four office rooms in a public building, 131a, 131b, 335, and 145b, [73], were associated with indoor-air-related health symptoms and abandoned by their occupants. One office room, 146, was provided with an air cleaner. The building and a floor plan showing the investigated rooms are described in Section 4.1. A building inspection revealed that the outdoor wall outside the problematic rooms was damaged and that rainwater had penetrated into the wall structure. A cork liner used as isolation inside the plinth in the outer wall was moist and degraded by microbes. Stereomicroscopic inspection of the liner indicated the growth of molds including *Aspergillus*, *Chaetomium*, and *Trichoderma* and an unrecognized fungus, as shown in Figure 1. Mineral wool insulation inside the outer wall was also moist and contaminated with mold. The inner surfaces of the five rooms and the collected samples of hard boards and gypsum liners exhibited no visible water damage or mold growth.

### 2.2. Diversity Tracking of Molds Cultivated from Wall Structures

The massive microbial growths cultivated on pieces of the moist cork liner (samples 1P61, 1K, 1POB) and mineral wool (sample 3MW) collected from outside the problematic rooms are shown in Figure 2. The major fungal colony types obtained on malt extract agar were light brown-white colonies (Figure 2A,I,J), green colonies (Figure 2B,G), green mycoparasitic colonies (Figure 2C,D), yellow and black colonies (Figure 2E,F), and gray yellow-green colonies (Figure 2G,H). An odd antifungal colony that presented only on single plates is shown in Figure 2K. Figure 2L shows the major bacterial colonizer of the cork liner cultivated on tryptic soy agar (spore-forming actinobacterium).

The diversity of the major fungal isolates was characterized as follows: biomass lysates of five colonies from each plate were tested for toxic responses by two rapid screening bioassays, boar sperm motility inhibition (BSMI) assay and inhibition of cell proliferation (ICP), and fluorescence emission. These bioassays showed that more than 70% of the tested colonies were toxic. Conidiophores and conidia/spores were inspected using a phase contrast microscope. The toxigenic colonies from plates A to K (Figure 2) were grouped into 10 morphotypes as shown in Table 1: brown-white colonies with *Acrostalagmus*-like conidiophores (MT1), tree *Aspergillus* morphotypes differing in fluorescence emission and toxic response (MT2, MT3, MT4), two *Trichoderma* morphotypes differing in conidia size (MT5, MT6)*,* and one toxigenic *Penicillium* morphotype (MT7). The final morphotype was obtained by microscopic inspection of the cork liner, representing a potentially toxic ascomata-producing *Chaetomium-*like morphotype (MT11). The major bacterial colonizers were spore-forming actinobacteria that were toxic in either BSMI or ICP or both (MT8-MT10).

Representatives of morphotypes MT1 to MT7 were identified by ITS sequencing as *Acrostalagmus luteoalbus* (MT1)*, Trichoderma atroviride* (MT5), *Trichoderma trixiae* (MT6), and *Penicillium expansum* (MT7). The isolates assigned to morphotypes MT2, MT3, and MT4 were identified as belonging to *Aspergillus* section *Versicolores,* section *Circumdati*, and section *Nigri*, respectively, based on similarity to the reference strains SL/3, PP2, and HAMBI-1271. The spore-forming actinobacteria were morphologically identical to each other but were separated into three morphotypes, MT8-MT10, based on their toxic response in the bioassays.

### 2.3. Cultured Settled Dust from Problematic Rooms Revealed Five Major Morphotypes

We looked for representatives of the genera colonizing the cork liner and mineral wool, *Acrostalagmus, Aspergillus, Trichoderma, Penicillium*, and *Chaetomium,* in settled dust collected from problematic rooms 131a, 131b, 335, and 145b, from nonproblematic rooms 134 and 223, and from room 146, where the occupant did not complain but had an air cleaner installed. The plates containing cultured settled dust are shown in Figure 3.

A majority (>70%) of the tested colonies in dust from the problematic rooms and room 146 showed a toxic response in the bioassays. The results in Table 2 show the toxigenic morphotypes in settled indoor dust: isolates similar to *Acrostalagmus* (MT1); *Aspergillu*s section *Versicolores* (MT2), *Circumdati* (MT3), and *Nigri* (MT4); mycoparasitic isolates similar to *T. trixiae* (MT5) and *T. atroviride* (MT6); and *P. expansum* (MT7) and *Chaetomium* isolates (MT11). One isolate, MH52, was identified by ITS sequencing as *Chaetomium globosum*. The results in Table 1 and Table 2 show that the toxigenic fungal morphotypes mainly found in the cork liner were also detected in the settled dust. Settled dust from room 146 looked similar to the dust shown in Figure 3G, containing mainly toxigenic *Chaetomium-*like colonies. Settled dust from rooms 223 and 134, located farther from the water-damaged site of the wall, contained mainly nontoxic *Rhizopus*; none of the 40 tested biomass dispersals were toxic in the bioassays, and the occurrence of toxic colonies was <5%.

### 2.4. Characterization of Acrostalagmus luteoalbus Strain POB8

The strains representing morphotype MT1, similar to *A. luteoalbus* strain POB8 shown in Figure 4, were isolated as a major colonizer from cork liner 1P61 and 1POB and from settled dust from room 131b.

Ethanol extracts prepared from plate-grown biomass of strains POB8, A1/K, A2/K, A3/K, A4/K, and POB1 exhibited similar blue fluorescence to the biomass dispersal shown in Table 1. The bioreactivity of extracts of biomass from the plate of strain POB8 shown in Figure 4 and the five strains representing the same morphotype was tested.

To reveal the diversity of the five *Acrostalagmus* sp. strains compared with *Acrostalagmus luteoalbus* POB8 isolated from the cork liners and indoor dust, they were tested with four complementary bioassays. The tests are described and referenced in Section 4.4.1 and Section 4.4.2. Briefly, the two BSMI assays measured sublethal toxicity as inhibition of motility in exposed sperm cells. The spermatozoa membrane integrity disruption (SMID) assay measured lethal toxicity as a loss of plasma membrane integrity. The ICP assay measured cytostatic toxicity as a loss of the proliferating ability of a somatic cell line, PK-15. The results in Table 3 enable a comparison of the EC_50_ concentrations for the toxicity endpoints from the assays measuring different biological activities. A comparison of the EC_50_ concentrations obtained in the four bioassays revealed a characteristic and uniform toxicity profile for the five *Acrostalagmus* sp. isolates and *A. luteoalbus* strain POB8. For the six *Acrostalagmus* strains, the toxic endpoints in terms of EC_50_ concentrations were similar in the three assays, around 10 µg mL^−1^, indicating that the *Acrostalagmus* extracts, in contrast to the extracts of reference strains, inhibited sperm motility and cell proliferation at the same concentrations after 1 and 3 d, respectively. The low toxic endpoints in the SMID assay also indicated a rapid lethal effect in sperm cells exposed at 37 °C. Briefly, the toxicity profile revealed by the bioassays indicated that the *Acrostalagmus* extracts exhibited lethal and cytostatic toxicity when exposed to cells at 37 °C and a motility-inhibiting effect when exposed to cells at 22 °C for 1 d. None of the *Acrostalagmus* strains exhibited a rapid motility-inhibiting effect after 20 min of exposure at 22 °C. This uniform toxicity profile of the six *Acrostalagmus* sp. isolates (A1/K, A2/K, A3/K, A4/K, and POB1) and *A. luteoalbus* POB8 separated them from the reference strains. This indicates that the strains may have produced the same bioactive metabolites, strengthening the hypothetical identity of the five *Acrostalagmus* sp. strains as *A. luteoalbus*.

Stereomicrographs of *A. luteoalbus* strain POB8 (Figure 4) show that the metabolically active biomass secreted exudates and vesicles. The toxic endpoints of the liquid exudate collected from strain POB8 exhibited a toxicity profile and blue fluorescence similar to those of the ethanol extracts. This indicates that the blue fluorescing exudates possibly contained the same substances as the blue fluorescing ethanol extracts. The liquid exudates of the reference strains of genus *Aspergillus* exhibited no toxicity in the bioassays, whereas exudates of the *Penicillium, Stachybotrys*, and *Chaetomium* strains secreted toxins in their guttation droplets and/or exudates.

### 2.5. Compounds of Toxic A. luteoalbus POB8 Ethanol Extract Were Identified as Melinacidins II, III, and IV

The ethanol extract of *A. luteoalbus* POB8 was analyzed using high-performance liquid chromatography–electrospray ionization mass spectrometry (HPLC-ESI-MS) (Figure 5A–D). On the HPLC-MS total ion chromatograms of the ethanol extract of *A. luteoalbus*, three compounds were identified, as shown in Figure 5A. Compound **1**, with a retention time of 8.5 min, had a protonated mass ion [M + H]^+^ at *m/z* 729.2, a sodiated mass ion [M + Na]^+^ at *m/z* 751.2, and a sodiated mass ion of dimer [2M + Na]^+^ at *m/z* 1479.0 (Figure 5B). Compound **2**, with a retention time of 12.2 min, had a protonated mass ion [M + H]^+^ at *m/z* 713.2, a sodiated mass ion [M + Na]^+^ at *m/z* 735.2, and a sodiated mass ion of dimer [2M + Na]^+^ at *m/z* 1447.0 (Figure 5C). Compound **3**, with a retention time of 17.7 min, had a protonated mass ion [M + H]^+^ at *m/z* 697.2, a sodiated mass ion [M + Na]^+^ at *m/z* 719.2, and a sodiated mass ion of dimer [2M + Na]^+^ at *m/z* 1415.0, with retention time of 17.7 min (Figure 5D). The obtained mass spectrometry data of compounds **1**–**3** matched the dimeric epipolythiodioxopiperazines melinacidin IV (728 g/mol), III (712 g/mol), and II (696 g/mol), respectively, reported earlier [74,75]. 

The obtained MS/MS data of compounds **1**–**3** were similar to those reported for the dimeric epipolythiodioxopiperazines [76,77,78]. The MS/MS spectrum fragmentation patterns (Figure 6A) of compound **1** were identical to the reported MS/MS spectrum of melinacidin IV [76]. Furthermore, the mass spectra and fragmentation patterns of melinacidins IV, III, and II highly resemble each other (Figure 6A–C). According to the obtained and reported MS and MS/MS mass spectrometry data, compounds **1**, **2**, and **3** of *A. luteoalbus* were identified as melinacidin IV, III, and II, respectively. The amounts of melinacidin IV (230 µg mL^−1^), III (290 µg mL^−1^), and II (120 µg mL^−1^) in the ethanol extract of *A. luteoalbus* were calculated from the total absorbance (220 nm) of the HPLC-UV chromatograms.

## 3. Discussion

This article describes the detection of cultivable toxigenic fungi in wet outdoor walls and indoor dust in a building with indoor air quality problems. One of the species found, melinacidin-producing *Acrostalagmus luteoalbus*, was detected as building-associated mold for the first time. This study illustrated that the mixed mycobiota cultivated from the wet wall and from settled indoor dust in problematic rooms seemed to contain the same major fungal species and genera. This indicates a possible connection between outdoor structures and indoor spaces (Figure 1, Figure 2 and Figure 3, Table 1 and Table 2).

Interestingly, a rare mold genus identified as *Acrostalagmus*, which was the major constituent of the mixed mycobiota in the wet cork liner from the water-damaged outdoor wall, was also present in indoor dust. One representative strain, POB8, was identified at the species level as *A. luteoalbus* by ITS sequencing. This strain has been used as a reference strain [65,66,79,80], but otherwise there are no reports of *Acrostalagmus* sp. strains growing on building materials, globally or in Finland. Based on its distinct morphology and phylogenetic distance, *A. luteoalbus* (basionym *Sporotrichum luteoalbum*) was introduced as a generic distinction between the former sections *Verticillium* and *Nigrescentia* [81,82,83,84]. The occurrence of *Verticillium* in building materials and cork has been described, but no isolates have been identified as *Acrostalagmus* species [10,85]. The production of immunosuppressive and cytotoxic melinacidin by an *A. luteoalbus* isolate has been shown [74,86]. The detected bioreactivity in biomass and guttation droplets indicate the possibility of a bioreactive microbial secretome including melinacidins migrating from a wet outdoor wall into indoor spaces.

This study presents the mass spectrometry data characteristics of melinacidins and shows that *A. luteoalbu*s strain POB8 produces melinacidins II, III, and IV. Melinacidin derivatives were previously reported from a variety of fungi including *Acrostalagmus luteoalbus* (syn. *A. cinnabarinus*) [74,86]. Melinacidins belonging to the epipolythiodioxopiperazines (ETPs) are related but not identical to verticillin, chaetocin, and gliotoxin, the best-known ETPs produced by *Aspergillus fumigatus.* Epipolythiodioxopiperazines have antiproliferative, cytotoxic, immunomodulatory, antiviral, and antimicrobial activity in vitro [77,87] and have been shown to be toxic to mammals; the LD in mice (i.p.) is 2–4 mg/kg [75,77,88,89,90]. The toxicity of ETPs depends on a disulfide bridge, with inactivating enzymes as methyl transferases via reactions with thiol groups. They also generate reactive oxygen species by redox cycling, inducing oxidative stress and mitochondrial damage [88,91]. This may explain the newly observed toxicity to sperm cells. However, this has yet to be confirmed with pure melinacidins.

Assuming that melinacidins were the only toxic substances in the ethanol extracts, the EC_50_ concentration in the BMSI and ICP assays for the tested melinacidin mixture was calculated as 0.3 to 0.6 µg mL^−1^. Based on a similar toxicity profile and blue fluorescence, it is possible, but not proven, that the liquid exudates secreted from the growing biomass of POB8 (Figure 4B) also contained melinacidins. Melinacidin concentration in the exudate, when calculated based on the toxic response, would be around 40 to 80 µg melinacidins per mL exudate. The hypothetically calculated melinacidin content in the biomass (wet wt) and exudates would be around 0.8 to 2 µg and 0.4 to 0.8 µg melinacidins mg^−1^, respectively. This means that similar amounts of melinacidins could migrate with the exudate as with fungal particles.

Four other genera were found in both the wet outdoor wall and the indoor dust from the three problematic rooms in the old wing of the building (and room 146 with the air cleaner) close to the wet wall, but not from dust from the two more remote nonproblematic rooms. The fungi representing the genera *Aspergillus, Trichoderma*, *Penicillium*, and *Chaetomium* are common constituents of mycotoxin-producing indoor mycobiota and common colonizers of indoor building materials [1,65,66,72,79,92,93]. These fungi are also listed as indicator species for water damage [1,94,95].

Occupants in the rooms complained about indoor air quality even though no signs of microbial growth or water damage were detected, and the airborne load of microbial particles (<4 CFU m^−3^) was too low to explain the reported symptoms. However, the negative pressure of 3–4 Pa and a sealing repair that was performed led to the suspicion of air leakage through the building structure [1,73]. Although microbial contamination inside buildings does not necessarily have direct contact with the indoor air, microbial growth within the exterior walls can affect indoor air quality. This can happen if, as a consequence of fluctuations in wind and indoor air pressure, the infiltration airflow drifts through a contaminated wall structure [96,97,98,99,100].

The occurrence of the same molds in the wet wall and indoor dust indicates possible transmission of conidia and spores into the problematic rooms and enrichment in settled dust. Tiny, potentially antibiotic-producing spores of *Streptomyces* migrated from the construction material into the indoor air [96]. In addition to conidia and fragments of microorganisms, the substances of microbial metabolism [6,72,79], secreted in liquids in guttation droplets and vesicles, may be trapped within building insulation and structural elements. The fungal secretome and fungal metabolites (including proteins, peptides, surface active substances, mycotoxins, VOCs, etc.) may be transported to interior spaces via liquid and vapor fluxes within materials and via airflow (negative pressure) within ventilation systems and rooms [1,96,97,98,99,100,101,102,103,104,105]. The microbial liquid metabolites, secretome, and VOC emissions [106,107] may enhance and exceed the indoor concentrations of bioreactive agents provided by airborne conidia and fragments [9,13,65,68,79]. Fungal protein homologues of human proteins that initiate or signal tissue damage, mycotoxins, and microbial mitochondrial toxins have all been reported to enhance and provoke inflammatory responses [103,104,105]. Indoor microbes may produce and emit airborne surface-active agents, enhancing the immunoreactivity of inhaled allergens [6,106,107]. The occupants of the problematic rooms may have been exposed to indoor air polluted with immunoreactive metabolites and surface-active agents from microbial species actively growing on the cork liner and mineral wool in the wet outdoor wall. This exposure combined with a low tolerance to environmental microbes may have decreased their resilience and would explain the symptoms they experienced. The noncomplaining occupant in room 146 may have been more resilient to the exposure, or the air in the room may have been efficiently cleaned by the air cleaner.

In conclusion, we suggest that the outdoor wall was the potential emission source for the indoor mold contamination detected in the problematic rooms. In addition to viable conidia, the mixed microbiota colonizing the wet outer wall possibly also emitted immunoreactive microbial metabolites among the exudates, vesicles, and fungal fragments into indoor air exposing the occupants of the problematic rooms. We also suggest that exposure to the immunoreactive metabolites may have attenuated the immunological tolerance to commensal and environmental microbes. There are no methods yet for measuring the total load of airborne immunoreactive fungal metabolites in indoor air, or for measuring decreased resilience to potentially hazardous exposure. Our proposed connection in this study between microbial emissions from the wet outer wall and the reported complaints concerning the indoor air quality is speculative but cannot be excluded.

## 4. Materials and Methods 

### 4.1. A Public Building from the 1960s Investigated for Mold Contamination

A public building in southern Finland that was involved in indoor-air-related complaints was investigated for mold contamination during 2013–2014, as described in [73]. The building consisted of two parts: an old wing built in 1957 (rooms 145b, 146, and 335) and a new wing erected in 1963 (rooms 131a, 131b, 223, and 134); it was concrete-framed and brick-lined and had mineral wool as isolation material. The building still contained the original construction materials, such as a cork liner in the building network (Figure 7 and Figure 8), and mechanical ventilation was installed in the 1970s. Ventilation was turned off from 8:00 pm to 6:00 am on workdays and during weekends. The floor plan of the building, the sites of the investigated rooms, and the structure of the outer wall are shown in Figure 1. The building underwent an indoor air survey in 2010 and problematic rooms were abandoned. Reported symptoms included respiratory distress, increased oxygen demand, the need for a portable oxygen cylinder, and skin symptoms (personal communication from the occupants).

A new survey of rooms 131a and 131b early in 2013 revealed a negative pressure of 3–4 Pa in relation to outdoor pressure, so sealing repair was performed. No elevated concentrations of airborne microbes (<4 cfu m^3^) were detected, but the indoor air quality complaints continued. Samples had been taken from the rooms by methods approved for official use in Finland for health risk assessment of indoor air, i.e., VOC and airborne microbes with the Andersen impactor [108,109]. The results obtained by analyzing these samples did not indicate the causative agents for the symptoms experienced by the occupants or explain their complaints. 

The rooms classified as problematic remained abandoned. After renovation and removal of the insulation material in the outer wall in 2016–2018, rooms 131a and 145b were returned to ordinary use, whereas rooms 131b and 335 were used for equipment storage.

### 4.2. Collection of Samples and Diversity Tracking of Microbial Constituents in Building Materials and Settled Dust

The sampling sites and the degraded cork liner colonized by microbes are shown in Figure 7 and 8. The samples of the building network were taken by drilling with a shock drill through the wall (Figure 8A). When the drill penetrated the concrete, the concrete dust was removed by vacuuming. Then, a tube was fixed to the drill (Figure 8B) and pressed into the cork liner, filling the tube with cork. The cork trapped in the tube was aseptically removed into sterile plastic bags, as shown in Figure 8C.

Mineral wool was collected from inside the wall on the third floor (Figure 7), and pieces of hard board, gypsum liners, and settled dust from inner surfaces 1–2 m above floor level in the problematic rooms were collected into sterile plastic bags. The material samples were inspected with a stereomicroscope and fluorescence microscope, using 400× magnification (Nikon Eclipse E600, Nikon Corporation, Tokyo, Japan) with BP330-380 nm/LP400 nm filters, and stored at −20 °C before cultivation. Staining with fluorescent Hoechst and propidium iodide stains (Figure 8A–C) was described in Andersson et al. [6].

### 4.3. Experimental Design for Cultivation and Identification of Microbial Isolates

The methods used for cultivating the mold colonies were described in [6]. Pieces of the material samples, about 2 mm × 2 mm × 2 mm, were rubbed on the surface of malt extract agar plates (15 g malt extract from Sharlab, Barcelona, Spain, and 12 g of agar from Amresco, Solon, OH, USA, in 500 mL of H_2_O) and on tryptic soy agar plates (Sharlab, Barcelona, Spain). Settled dust collected on cotton swabs was streaked on malt extract agar. Plates were incubated for 4 weeks at 22–24 °C.

A scheme illustrating the experimental design for tracking the diversity of major microbial constituents in building material and dust is shown in Figure 9.

Separation of the isolates into morphotypes (MTs) was based on the toxicity profile and morphology of conidiophores, conidia, ascomata, and ascospores obtained with a phase-contrast microscope (400× magnification; Olympus CKX41, Tokyo, Japan) and image recording software (cellSens^®^ standard v. 11.0.06, 2012, Olympus Soft Imaging Solutions GmbH, Münster, Germany) and compared to reference strains according to Samson [110,111] as described in [65,66]. 

Selected representatives of the morphotypes were identified in previous studies by ITS or *tef1α* sequencing with the primer pairs ITS1 (5′-TCCGTAGGTGAACCTGCGG-3′)/ITS4 (5′-TCCTCCGCTTATTGATATGC-3′) and EF595F (5′-CGTGACTTCATCAAGAAGATG-3′)/EF1160R (5′-CCGATCTTGTAGACGTCCTG-3′), respectively [79].

### 4.4. Toxicity Assays

#### 4.4.1. Rapid Screening Test of Single Colonies

Rapid screening tests applied directly to the primary sampling plates to measure (a) the toxins affecting the cellular energy metabolism, mitochondria, and ion homeostasis based on inhibition of boar spermatozoa motility (BSMI), and (b) the toxins affecting macromolecular synthesis and cytostatic activity based on inhibition of the proliferation of somatic cell line PK-15 (ICP) were previously described in detail [65,79]. In this study, the colony biomass was suspended in 200 μL of ethanol and heated in a water bath to 55–60 °C for 10 min. A colony was considered very toxic in the BSMI assay when <2.5 vol.% of its biomass suspension inhibited boar sperm motility after 30 min to 1 day of exposure, and slightly toxic if motility inhibition occurred after 3 days of exposure. A colony was considered toxic in the in vitro ICP assay when <5 vol.% resulted in inhibition of cell proliferation of porcine kidney (PK-15) cells after 2 days of exposure in ICP assays.

#### 4.4.2. Toxicity Assays for Ethanol-Extracted Pure Fungal Cultures

Toxicity assays involving the ethanol extraction of lipophilic bioactive peptides and mycotoxins [22,31,65,66,80,93] obtained from pure fungal cultures were performed using porcine cells (sperm and somatic cell line PK-15) as indicators according to previously described methods [79,80]. The toxic response in the bioassays was measured for the ethanol extracts as toxic endpoints defined as EC_50_ concentrations, i.e., the lowest concentration of ethanol-soluble dry substances per mL of target cell suspension causing an adverse (toxic) effect in 50% of exposed cells. The test procedures and calculation of EC_50_ for the ethanol-dry substances and pure mycotoxins in the BSMI, SMID, and ICP assays were described previously [79,80,112]. The BSMI assay measured sublethal toxicity as disturbance in the mitochondrial activity, ion homeostasis, and energy supply of the exposed sperm cells. The BSMI_M_ and BSMI_R_ assays measured motility after exposure of motile and resting spermatozoa, respectively. The SMID assay measured lethal toxicity as loss of plasma membrane integrity. The ICP assay measured cytostatic toxicity and cell death as the loss of the proliferating ability of growing somatic cells, which may be caused by inhibition of macromolecular synthesis and/or induction of necrotic or apoptotic cell death [65,66,73,80]. 

### 4.5. Chemical Analysis

The biomass of *A. luteoalbus* strain POB8 was harvested from a malt extract agar (MEA) plate incubated at room temperature for 10 days. The collected biomass, about 200–400 mg wet weight of *A. luteoalbus*, was extracted with ethanol, and the toxic ethanolic extract was analyzed by high-performance liquid chromatography–ion trap mass spectrometry (HPLC-IT-MS) as described by Salo et al. [65]. HPLC–electrospray ionization ion trap mass spectrometry analysis (ESI-IT-MS) was performed using an MSD-Trap-XCT plus ion trap mass spectrometer equipped with an Agilent ESI source and Agilent 1100 series LC (Agilent Technologies, Wilmington, DE, USA) in positive mode with a mass range of *m/z* 50–2000. ESI source parameters used for analysis were: nebulizer gas pressure, 35 psi; drying gas flow rate, 8 L min^−1^; drying gas temperature, 350 °C; and capillary voltage, 5000 V. The column used was a SunFire C18, 2.1 × 50 mm, 2.5 μm (Waters, Milford, MA, USA). Separation of compounds from the ethanol extract of biomass of *A. luteoalbus* strain was done using an isocratic method of solution A, H_2_O with 0.1% (*v*/*v*) formic acid, and B, methanol in a ratio of 40/60 (*v*/*v*), for 15 min and a gradient of 100% B for 50 min at a flow rate of 0.2 mL min^−1^. Identification of compounds of the toxic ethanolic extract was done by HPLC-IT-MS and MS/MS analysis. The amounts of identified compounds were calculated from total absorbance (220 nm) of the HPLC-UV chromatogram of the *A. luteoalbus* POB8 ethanol-extracted biomass.

## Figures and Tables

**Figure 1 pathogens-10-00843-f001:**
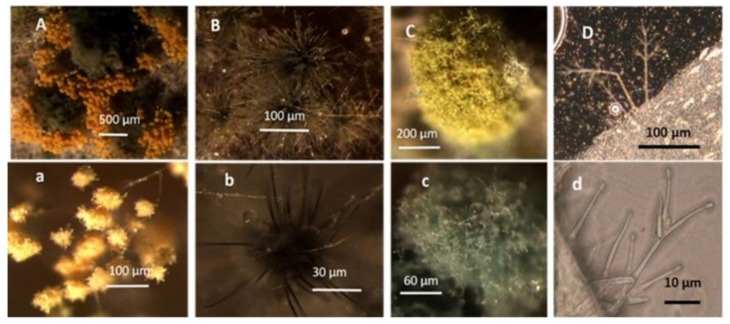
Stereomicrographs of major fungi colonizing moist cork liner. (**A,a**) *Aspergillus*-like conidiophores; (**B,b**) ascomata characteristic for *Chaetomium*-like fungi; (**C,c**) mycoparasitic *Trichoderma*-like colonies; and (**D,d**) a dominant colonizer, a fungus with straight erect repeatedly branched orange conidiophores, 100 × 4–4.5 μm.

**Figure 2 pathogens-10-00843-f002:**
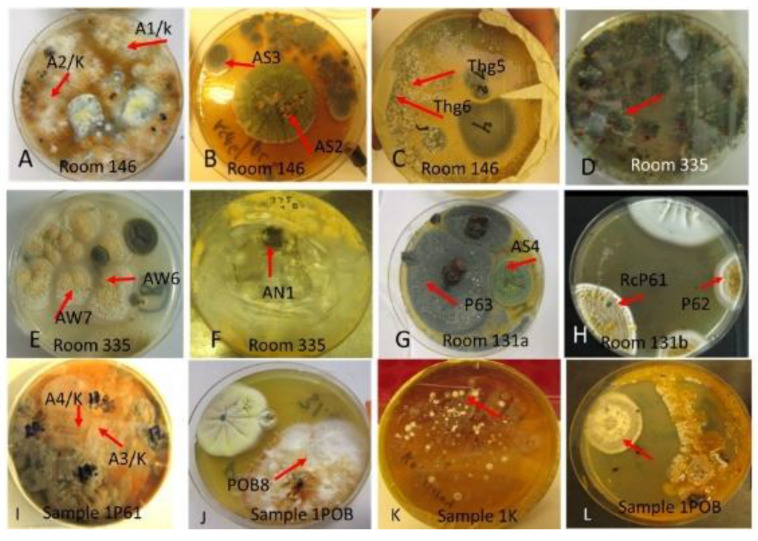
Pieces of moist cork liner and mineral wool cultivated on malt extract agar for 14 d at 22 °C. (**A**–**C**,**G**,**H**); dominant colonies from cork liner (samples 1P61,1POB,1K). (**D**–**F**) Major colonies from mineral wool (sample 3MW). Origin of isolates characterized/identified are marked with red arrows and representative strain codes. (**K**) Colonies of white, spore-forming actinobacteria; (**L**) unidentified toxigenic and antagonistic fungicidic colony type appearing on this plate only.

**Figure 3 pathogens-10-00843-f003:**
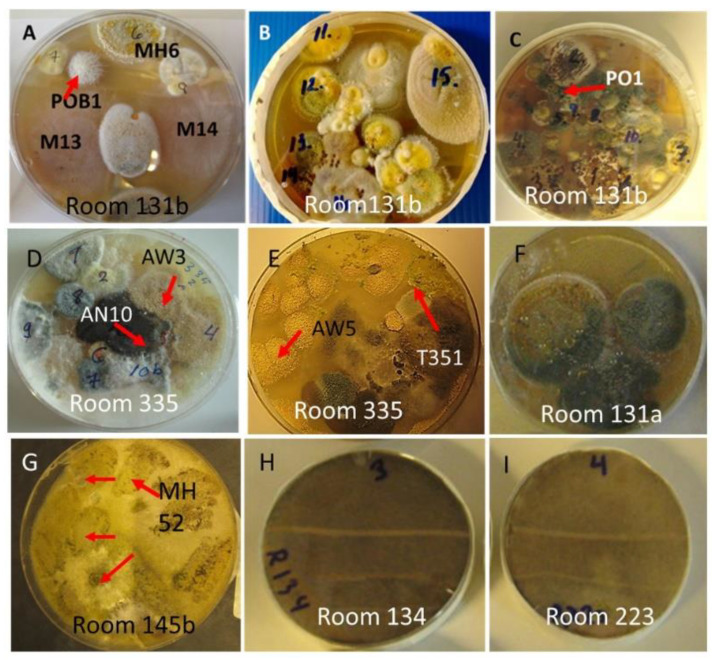
Settled dust collected from six rooms cultivated on MEA for 14 d. Origin of characterized/identified isolates are marked with red arrows and representative strain codes. (**A**–**G**) Major fungal colonies from dust collected from problematic rooms. (**H,I**) Dust cultivated from nonproblematic rooms in ordinary use.

**Figure 4 pathogens-10-00843-f004:**
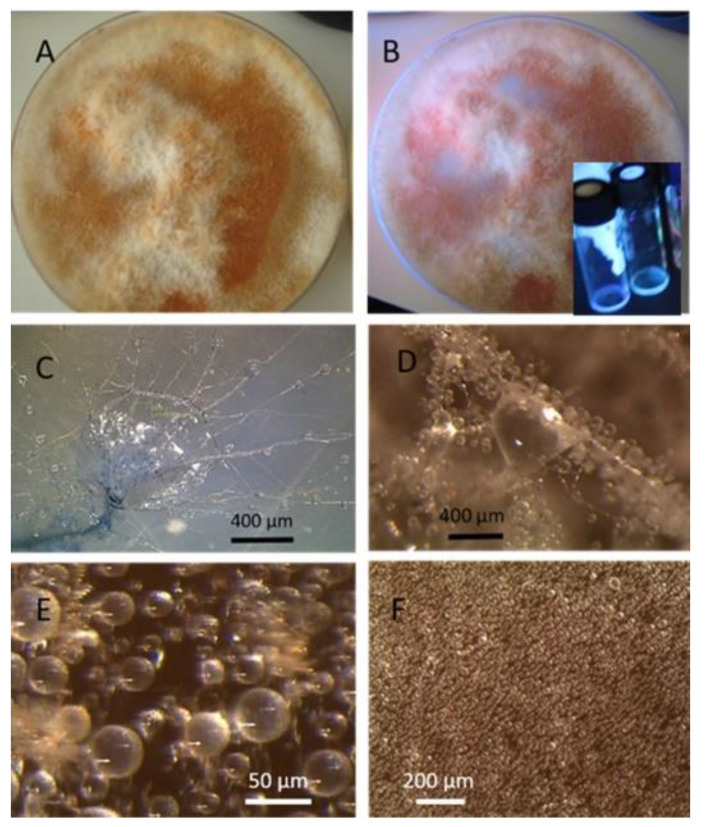
*Acrostalagmus luteoalbus* strain POB8 cultivated on malt extract agar for 10 d. (**A**) Pure-cultured strain in visible light. (**B**) Same plate exhibiting blue fluorescing exudates when excited with UV light (insert). (**C**–**F**) Stereomicroscopic views of plate in (**A**): (**C**,**D**) Exudates emitted from fungal hyphae. (**E**,**F**) Liberated vesicles.

**Figure 5 pathogens-10-00843-f005:**
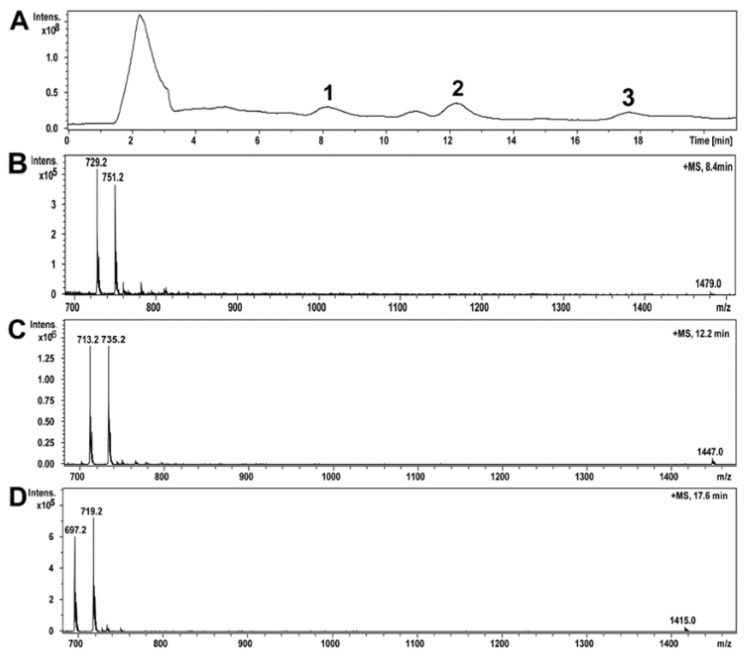
MS spectra of compounds of *Acrostalagmus luteoalbus* POB8 ethanol extract. (**A**) TIC of three compounds at retention times of 8.4 (1), 12.2 (2), and 17.8 min (3). (**B**–**D**) MS spectra of compounds **1**, **2**, and **3** had protonated mass ions at *m/z* 729.2, 713.2, and 697.2; sodiated mass ions at *m/z* 751.2, 735.2, and 719.2; and sodiated mass ions of dimers at *m/z* 1479.0, 1447.0, and 1415.0, respectively.

**Figure 6 pathogens-10-00843-f006:**
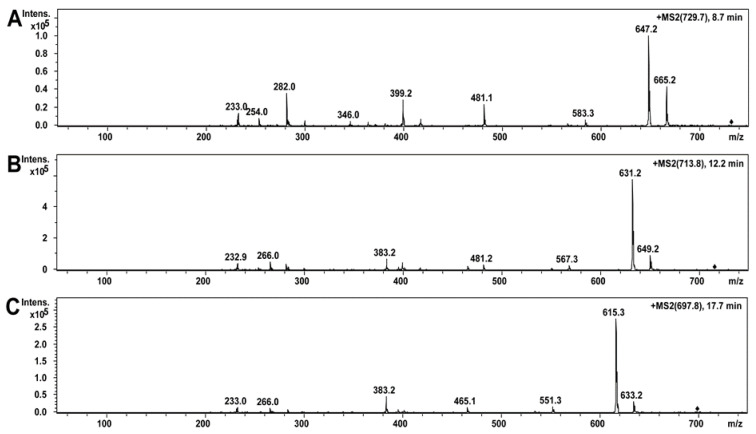
MS/MS spectra of compounds of *A. luteoalbus* POB8 ethanol extract: (**A**) compound **1** precursor ion *m/z* at 729.7; (**B**) compound **2** precursor ion *m/z* at 713.8; (**C**) compound **3** precursor ion *m/z* at 697.8.

**Figure 7 pathogens-10-00843-f007:**
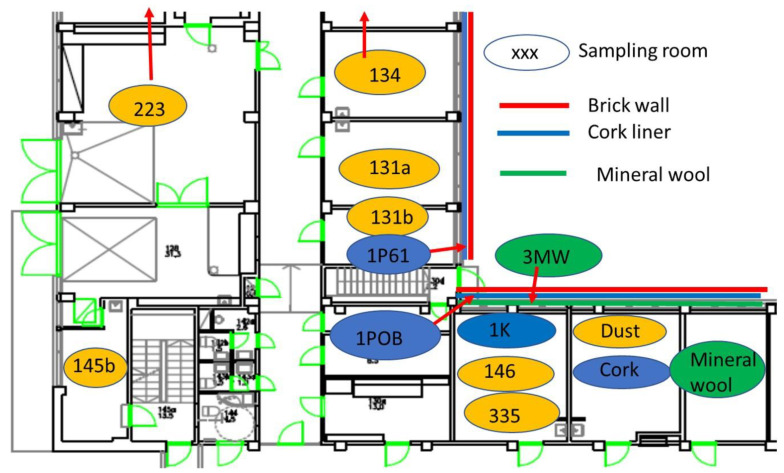
Floor plan of the public building investigated for mold growth. Samples of settled indoor dust and pieces of cork liner used as insulation inside plinth and mineral wool inside the outer brick wall were collected from selected rooms. The first number (1 to 3) in the sample code indicates the floor (first, second, third floor). Rooms 131a, 131b, and 335 were abandoned in 2010. Occupants in room 145b complained about indoor air and reported symptoms. Occupants in room 146 did not report symptoms until 2014 but had an air cleaner installed in the room. Rooms 134 and 223 were nonproblematic and in ordinary use.

**Figure 8 pathogens-10-00843-f008:**
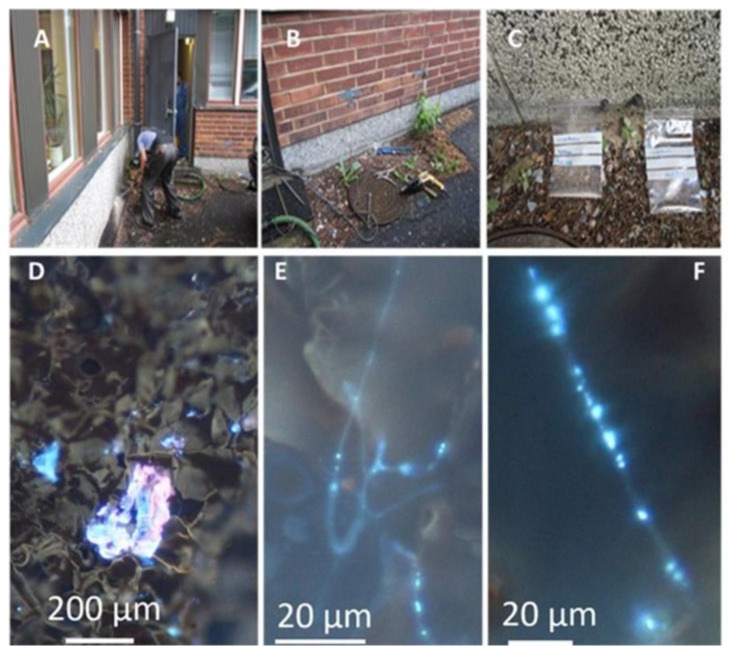
Sampling of building materials from water-damaged outdoor wall (upper panels), and micrographs of obtained samples (lower panels). Upper panels show (**A**) drilling of holes in plinth of outer wall; (**B**) used tools; and (**C**) obtained samples of wet cork liners stored in plastic bags. Lower panels show pieces of sampled wet cork liner with Hoechst 33342 nucleic acid stain + propidium iodide staining microbial structures containing DNA or RNA blue and red. (**D**) Degraded surface contaminated with microbes. (**E,F**) Growing actinobacterium and growing fungal hyphae.

**Figure 9 pathogens-10-00843-f009:**
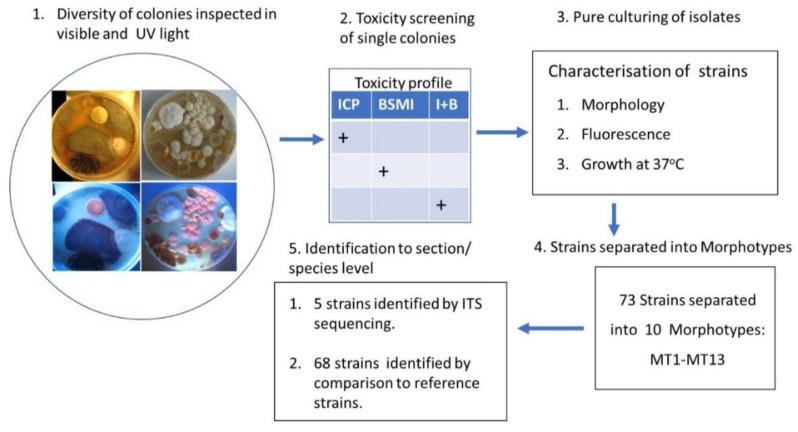
Scheme illustrating experimental design for tracking diversity of major microbial constituents in building materials and dust. After three weeks of incubation, colonies on primary isolation plates (not yet single-spored) were numbered and screened for toxicity. Toxic colonies were streaked pure, characterized, and separated into morphotypes. Representatives of morphotypes were identified by ITS sequencing or by comparison with reference strains according to [106].

**Table 1 pathogens-10-00843-t001:** Characterization of 38 isolates (morphotypes MT1-MT10) cultivated from pieces of cork liner and mineral wool from outdoor wall. Toxicity was measured with two bioassays, boar sperm motility inhibition (BSMI) and inhibition of proliferation (ICP) of porcine kidney cells (PK-15).

	Biomass Lysates	Morphology in Microscope	OriginSample
Morphotype	Toxicity *BSMI ICP	Fluorescence/Colony Color	Conidiophores	Conidia	
MT1	*Acrostalagmus luteoalbus*
Isolates: POB8 ^1^, A1/K, A2/K, A3/K, A4/K	−	+	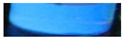 Brown/white	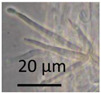	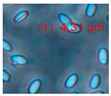	1P61, 1K
MT2	*Aspergillus* *section Versicolores*
Isolates: AS1, AS2 AS3, AS4, AS5	−	+	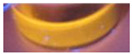 Green/yellow	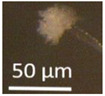	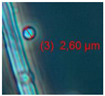	1POB 1P611K
MT2	Reference strain *Aspergillus versicolor* SL/3 ^6^
	−	+	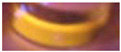 Green/yellow	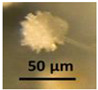	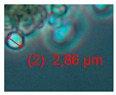	
MT3	*Aspergillus* section *Circumdati*
Isolates: AW7, AW8, AW9, AW10, AW11	+	+	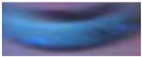 Yellow	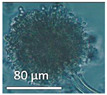	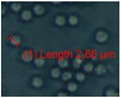	3MW
MT3	Reference strain *Aspergillus westerdijkiae* PP2 ^6^
			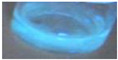 Yellow	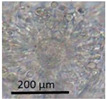	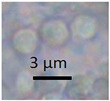	
MT4	*Aspergillus* section *Nigri*	
Isolates: AN1, AN2, AN3, AN7, AN11	−	+	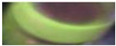 Black	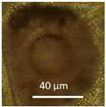	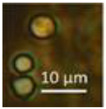	3MW1K
MT4	Reference strain *Aspergillus niger* HAMBI 1271
			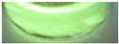 Black	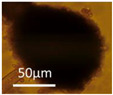	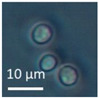	
MT5	*Trichoderma atroviride*
Isolates: Tri335 ^2^Tri335b, Tri335cTri336, Tri337	+	+	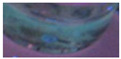 Green	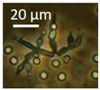	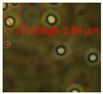	3MW
MT6	*Trichoderma trixiae*
Isolates: Th1/kg, Th2/kg, Th4/kg, Th5/kg ^3^, Th6/kg ^4^			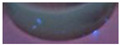 Green	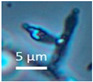	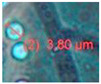	1K 3MW
MT7	*Penicillium expansum*
Isolates: Rc P61 ^5^, RcP62	(+)	+	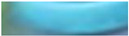 Green-yellow/gray	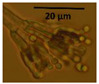	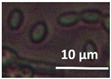	1P61
MT8–MT10	Spore-forming *Actinobacteria*
Isolates: Str3/KN, Str4/KN	+	+	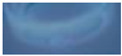	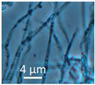	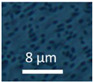	1K
Isolates: Str1/KN, Str2/KN	+		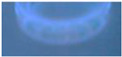	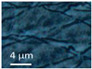	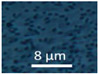	1K
Isolates: Str5/KN, Str6/KN		+	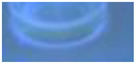	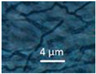	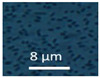	1K

+ Test with most toxic response, toxic endpoints of + < (+) < −. A colony was considered toxic in BSMI assay if ≤2.5 mg mL^−1^ of biomass inhibited sperm motility after 30 min (= +) on one day (= (+)) of exposure, and in ICP if ≤5 mg mL^−1^ of biomass inhibited proliferation of PK-15 cells exposed for 2 d. GenBank accession numbers: ^1^ KM853014 (ITS); ^2^MH176998 (*tef1α*); ^3^ MZ229302 (*tef1α*); ^4^ MZ229303 (*tef1α*); ^5^ MK201596 (ITS), KP889005 (*cmd*); ^6^ identified by DSMZ.

**Table 2 pathogens-10-00843-t002:** Characterization of 37 isolates from cultivated settled dust into nine morphotypes. Bioreactivity was tested as responses to two bioassays, boar sperm motility inhibition (BSMI) and inhibition of cell proliferation (ICP), with PK-15 cell line.

	Biomass Lysates	Morphology in Microscope	Origin
	Toxicity	Fluorescence/Colony Color	Conidiophores	Conidia	Room
Morphotype	BSMI	ICP	
MT1	*Acrostalagmus* sp.
Isolates: POB1, PH20	−	+	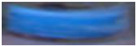 Brown/white	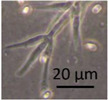	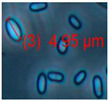	131b
MT2	*Aspergillus* section *Versicolores*
Isolates: A20, A11, A13, A14, A15	−	+	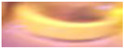 Green/yellow	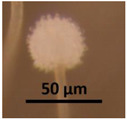	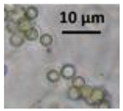	335131b
MT3	*Aspergillus* section *Circumdati*
Isolates: AW1, AW2, AW3, AW4, Aw5	+	+	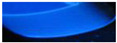 Yellow	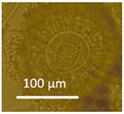	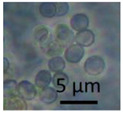	335
MT4	*Aspergillus* section *Nigri*
Isolates:AN10, AN11, AN12			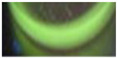 Black	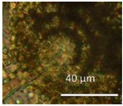	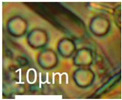	335145b
MT5	*Trichoderma* sp.
Isolates: PO1, PO2, PO3, PO4, PO5	+	+	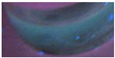 Green	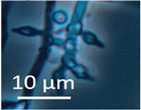	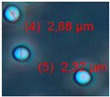	335
MT6	*Trichoderma* sp.
Isolates: T351, T355,T337, T338, T330	+	+	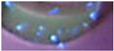 Green	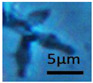	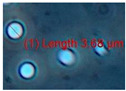	335
MT7	*Penicillium expansum*
Isolates: MH6 ^1^, P3a, P32, P33	(+)	+	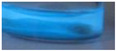 Green/yellow/gray	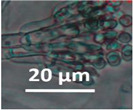	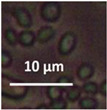	131b131a
MT11	*Chaetomium globosum*
Isolates: MH5, M13, M14, MH52 ^2^	+	+	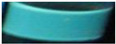 Black	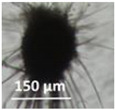	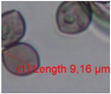	131b145b146
MT12	*Rhizopus* sp.
Isolates: R1, R2, R3, R4	−	−	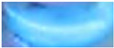 Black	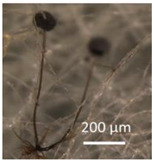	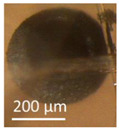	134223

+ Test with most toxic response, toxic endpoints of + < (+) < −. A colony was considered toxic in BSMI assay if ≤2.5 mg mL^−1^ of biomass inhibited sperm motility after 30 min (= +) on one day (= (+)) of exposure, and in ICP if ≤5 mg mL^−1^ of biomass inhibited proliferation of PK-15 cells exposed for 2 d. References: ^1^ [65], ^2^ [66].

**Table 3 pathogens-10-00843-t003:** Bioreactivity of ethanol-soluble substances and liquid exudates from plate-grown biomasses of strains identified as *Acrostalagmus luteoalbus**, Acrostalagmus* sp., and selected reference strains. Bioreactivity was measured as toxicity in four bioassays: boar sperm motility inhibition assay performed with motile and resting sperm cells (BSMI_M_, BSMI_R_) and inhibition of cell proliferation (ICP) with porcine kidney cell line PK-15.

Ethanol-Extracted Dry Substances		Exposure Concentrations EC_50_ µg mL^−1^		
	Boar sperm assays			Somatic cell line
	BSMI_R_	SMID_M_	BSM_R_	ICP
Exposure time and temperature	20 min, 22 °C	2 h, 37 °C	1 d, 22 °C	3 d, 37 °C
Strain code				
*Acrostalagmus luteoalbus* POB8	100	6	10	10
*Acrostalagmus* sp. Ac1/KG	100	6	10	10
*Acrostalagmus* sp. Ac2/kg	100	12	20	20
*Acrostalagmus* sp. AC3/kg	100	6	10	20
*Acrostalagmus* sp. AC4/kg	100	6	10	10
*Acrostalagmus* sp. POB1	100	12	20	10
Reference strains	Ethanol extracted dry substances		EC_50_ µg mL^−1^	
*Aspergillus versicolor* SL/3	>100	>100	20	1
*Chaetomium globosum* MTAV 35	ND	450	3	40
*Trichoderma atroviride* Tri335	5	2	5	60
Liquid exudates			Exposure concentration EC_50_ µL mL^−1^	
*Acrostalagmus luteoalbus* POB8	ND	ND	25	50
Reference strains	Liquid exudates		Exposure concentration EC_50_ µL mL^−1^	
*Aspergillus versicolor* SL/3	ND	ND	>100	>100
*Aspergillus westerdijkiae* PP2	ND	ND	>100	>100
*Aspergillus calidoustus* MH34	ND	ND	>100	>100
*Stachybotrys* sp. HJ5 *	ND	ND	>50	20
*Penicillium expansum* RCP61 *	ND	ND	2.5	0.04

BSMI_M_ and BSMI_R_, boar sperm motility inhibition assays performed with motile and resting sperm cells; SMID, sperm membrane integrity assay; ICP, inhibition of cell proliferation with porcine kidney cell line PK-15. * Presented in [65,66].

## Data Availability

Not applicable.

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
