# Peer review of "Melinacidin-Producing Acrostalagmus luteoalbus, a Major Constituent of Mixed Mycobiota Contaminating Insulation Material in an Outdoor Wall"

_pathogens, 2021, doi:10.3390/pathogens10070843_

Round 1

Reviewer 1 Report

The authors should consider the following(s):

  1. Please give rationales for using porcine kidney (PK-15) cells, instead of other cell models.
  2. How was the relative abundance of the proposed chemicals in the environment?
  3. Why did the author only choose ethanolic extracts? (other solvents, or successive extractions? what were the rationale?)
  4. Please provide a list of identified compounds in the article (or as supplementary information). DId the author use any reference chemical to check whether they identify the exact chemical? (or just using MS/MS patterns?)
  5. The authors should avoid over-stating /or over-extrapolate their findings, in abstracts and/or conclusion part.
  6. The authors may list the evidences of the secretome (or particular chemical) that posed actual effect to the occupants?
  7. The authors should provide details methdology of the LCMS performed
  8. Did the occupant or the chemical(s) would cause the nasal, lung irritation? any cellular or animal models study to support that? the relevant doses to cause detrimental effect?
  9. If causing skin irritation? any supporting model of 3D skin models? relevant animal model testings to support those?
  10. Regarding Indoor Air Quality, the authors may provide relevant questionnaires and detection of smell of the indoors. Please also list the relevance standards regarding those air requirement(s).
  11. The article may also discuss the outer wall cleaning policies and frequencies, relative humidity of the indoor and outdoors.
  12. Please give evidences of using BSMI (boar sperm motility inhibition assay), and their relevances to the research topics.
  13. Please provide full name of the abbreviation, i.e. SMID.
  14. The authors should highly consider using English proof-reading services by language professional.
  15. Please state clearly the novelty of this research in your abstract and conclusion.

Reviewer 2 Report

The manuscript describes the study of toxigenic species found in wet outside walls and indoor dust in the building with air quality problems. One of the species found, Acrostalagmus luteoalbus, was detected as building associated mold for the first time. The cytotoxic substances secreted by A. luteoalbus were identified as melinacidins II, III and IV. The manuscript contains novel data and has merit but the quality of presentation should be improved. I highly recommend to show the manuscript to the English native speaker or to use English editing service to improve stylistics and grammar. Also, there is a lot of misspellings, missed italics and etc.  

Please, check the Table 1 formatting. For several reference strains their morphotype codes are placed in the column with toxicity results.

Table 3 is formatted in a very confusing manner. It is hard to understand the relation of headings and the content of the columns. The data should be presented in clearer manner.

I’m not sure whether the term “secretome” can be used as it was done in L292-L322. Secretome in most cases means the secreted proteins, but many toxic substances secreted by toxigenic fungi are not proteins.

Figure 7 looks blurry and its quality should be improved for publication.   

Please, specify the chromatographic system which was used for analysis.

The Figure 5 caption is unnecessarily long and repeat the main text.

Please, specify why did you used these exact 5 toxicity assays for A. luteoalbus?  
